# *OsLRR-RLP2* Gene Regulates Immunity to *Magnaporthe oryzae* in *Japonica* Rice

**DOI:** 10.3390/ijms25042216

**Published:** 2024-02-12

**Authors:** Hyo-Jeong Kim, Jeong Woo Jang, Thuy Pham, Van Tuyet, Ji-Hyun Kim, Chan Woo Park, Yun-Shil Gho, Eui-Jung Kim, Soon-Wook Kwon, Jong-Seong Jeon, Sun Tae Kim, Ki-Hong Jung, Yu-Jin Kim

**Affiliations:** 1Department of Life Science and Environmental Biochemistry, Life and Industry Convergence Research Institute, Pusan National University, Miryang 50463, Republic of Korea; khjeong0115@naver.com (H.-J.K.); jihyun0_0@pusan.ac.kr (J.-H.K.); musgdmu8869@naver.com (C.W.P.); 2Department of Plant Bioscience, Life and Industry Convergence Research Institute, Pusan National University, Miryang 50463, Republic of Korea; com5222@naver.com (J.W.J.); swkwon@pusan.ac.kr (S.-W.K.); stkim5505@gmail.com (S.T.K.); 3Graduate School of Green Bio Science & Crop Biotech Institute, Kyung Hee University, Yongin 17104, Republic of Korea; phamthuy1107ls@gmail.com (T.P.); tuyetvankhu@gmail.com (V.T.); koyoong@khu.ac.kr (Y.-S.G.); alice804@khu.ac.kr (E.-J.K.); jjeon@khu.ac.kr (J.-S.J.)

**Keywords:** LRR-RLP, *Magnaporthe oryzae*, rice, *japonica* resistance

## Abstract

Rice is an important cereal crop worldwide, the growth of which is affected by rice blast disease, caused by the fungal pathogen *Magnaporthe oryzae*. As climate change increases the diversity of pathogens, the disease resistance genes (*R* genes) in plants must be identified. The major blast-resistance genes have been identified in *indica* rice varieties; therefore, *japonica* rice varieties with *R* genes now need to be identified. Because leucine-rich repeat (LRR) domain proteins possess *R*-gene properties, we used bioinformatics analysis to identify the rice candidate LRR domain receptor-like proteins (OsLRR-RLPs). *OsLRR-RLP2*, which contains six LRR domains, showed differences in the DNA sequence, containing 43 single-nucleotide polymorphisms (SNPs) in *indica* and *japonica* subpopulations. The results of the *M. oryzae* inoculation analysis indicated that *indica* varieties with partial deletion of *OsLRR-RLP2* showed susceptibility, whereas *japonica* varieties with intact *OsLRR-RLP2* showed resistance. The *oslrr-rlp2* mutant, generated using clustered regularly interspaced palindromic repeats (CRISPR)/CRISPR-associated protein 9 (Cas9), showed increased pathogen susceptibility, whereas plants overexpressing this gene showed pathogen resistance. These results indicate that *OsLRR-RLP2* confers resistance to rice, and *OsLRR-RLP2* may be useful for breeding resistant cultivars.

## 1. Introduction

Rice (*Oryza sativa* L.) accounts for approximately half of the total food crop consumption. However, global rice production is being seriously threatened by rice blast disease, which is caused by the fungus *Magnaporthe oryzae* [1]. Environmental conditions affect the active growth of *M. oryzae* and the blast disease develops at a relative humidity range of 92–96% and an air temperature range of 25–28 °C [2]. Rice blast diseases cause lesions on leaves, stems, peduncles, panicles, seeds, and roots [3]. Lesions can enlarge to several centimeters, eventually resulting in plant death [4]. The annual yield loss due to rice blast disease is approximately 10–30% [5]. Therefore, cultivating resistant varieties can reduce rice yield losses. Plants possess immune recognition pathways for pests and pathogens; resistance (*R*) genes provide disease resistance to plants. Many *R* genes are being studied; however, because of the increasing variety of pathogens and their resistance to *R* genes, additional research on *R* genes is important for rice breeding.

Plants defend themselves against pathogens through pattern-triggered immunity (PTI) and effector-triggered immunity (ETI) activation. The first plant defense mechanism is PTI, which is mediated by pattern recognition receptors (PRRs), which are activated upon direct recognition of pathogen/microbe-associated molecular patterns (PAMPs/MAMPs) by PRRs [6]. The second plant defense mechanism is ETI, which is mediated by the direct or indirect recognition of effectors by nucleotide-binding and leucine-rich repeat receptors (NLRs) [7]. The innate immune receptors can be initially recognized using NLR and PRRs. PRRs are located at the plasma membrane and recognize extracellular signals, termed PAMPs [8]. Receptor-like kinases (RLKs) are members of the PRR family. Receptor kinases perform recognition and signal transduction [9]. The intracellular kinase domains encoded by PRRs perform immune-sensing functions. In contrast to RLKs, receptor-like proteins (RLPs) contain an ectodomain, a transmembrane domain, and a short cytoplasmic tail [10]. As RLKs, RLPs are involved in plant defense and development [11,12]. Both use extracellular leucine-rich repeat (LRR) domains to perceive PAMPs; however, unlike RLKs, RLPs lack an intracellular kinase domain [13]. The molecular mechanisms underlying the recognition and signaling of RLPs are still poorly understood compared to RLKs [14]. Given the limited research available on this topic, further studies on receptor identification are required.

The LRR is involved in protein–protein recognition. The LRRs are generally 20–29 residues long and contain 11 conserved residues with the consensus sequence LxxLxLxxN/CxL, in which any amino acid is “x”; “L” is Val, Leu, or Ile, “N” is Asn, Thr, Ser, or Cys; and “C” is Cys or Ser [15]. Leucine-rich repeat receptor-like proteins (LRR-RLPs) are also involved in the activation of the immune response [16]. LRR-RLPs have been discovered in various plant species since they were first characterized in tomatoes (*Lycopersicon esculentum*) [11]. A total of 57 genes encoding LRR-RLPs have been identified in Arabidopsis [16], some of which have been functionally characterized [10]. For example, the receptor of the enigmatic MAMP of *Xanthomonas* (ReMAX) is responsible for the recognition of eMAX by *Xanthomonas* strains [17]. *Solanum lycopersicum* LRR-RLPs confer resistance against pathogens by recognizing avirulence proteins [18]. Ninety genes encoding LRR-RLPs have been identified [15]. OsRLP1 is involved in resistance to rice black-streaked dwarf virus [19]. RLPs have rarely been studied; research, particularly in rice, is scarce.

Cultivated rice varieties belong to the *indica* and *japonica* subspecies, which have substantially changed, including their stress resistance, during domestication [20,21]. However, most blast genes have been identified as *indica* accessions and are less well known in *japonica* cultivars [22]. In this study, we identified another RLP, named *OsLRR-RLP2*, in a *japonica* cultivar and evaluated its response to *M. oryzae*. To identify the *R* gene related to rice blast disease in the rice variety, we systematically analyzed single-nucleotide polymorphisms (SNPs) with indels and screened T-DNA insertional lines screening of the LRR domain protein genes. We obtained accessions possessing mutations within the LRR and subsequently performed a series of experimental investigations under blast treatment. Identification of the disease resistance genes related to PRRs provides genetic resources that can be used to improve our understanding of rice immunity. These results provide a useful reference for the development of rice plant with enhanced resistance to blast *R* gene.

## 2. Results

### 2.1. Selection of LRR Candidates and Phylogenetic Analysis

We first downloaded 3240 genes encoding LRR proteins from a rice genome database (Rice Genome Annotation Project, http://rice.uga.edu/, accessed on 21 October 2021). After removing duplicate values and changing them to the RAP-DB ID (https://rapdb.dna.affrc.go.jp/, accessed on 28 October 2021), 857 genes were analyzed for indels (≥3 bp) of 453 high quality rice varieties compared to *Nipponbare* and the frequency of indels was presented as a heatmap (yellow boxes indicate the high frequency and the blue ones indicates low frequency) (Figure 1A). We then identified 316 genes that showed high genetic diversity among the rice varieties compared to the reference (*Nipponbare*) genome. We further analyzed the microarray data for 277 genes with probes in rice Affymetrix gene chip (Figure 1B), and 16 genes of them showed significant changes in both the BLAST or bacterial blight disease treatments (Figure 1C). To verify the correlation between 16 genes and blast disease, we screened T-DNA-tagged mutant lines and conducted PCR-based genotyping for identification of the tagging in each gene. Through the analysis of the T-DNA mutant lines, one LRR domain gene was selected for further analysis due to its response to rice blast disease. This LRR domain gene, named *OsLRR-RLP2*, contains LRR N-terminus domain (LRRNT) and six LRR domains (Figure 2A).

To determine phylogenetic relationships with other species, the protein sequences of all species were searched using BLASTx in NCBI GenBank. We found high amino acid sequence identity with *Oryza* species: 93% similarity with *O. sativa indica* sequence (EAY86640.1) and 83.1% similarity with *Oryza glaberrima* (XP_052143053.1) (Figure 2B). Other species, including monocot species, have low amino acid sequence identities ranging from 39% to 55%. To find homologs in the model plant Arabidopsis, we also searched for homologs in the TAIR database (TAIR https://www.arabidopsis.org/, accessed on 13 June 2022). Although their similarity was very low (39.4%), the homolog with the highest similarity, AtRLP19 (OAP10192.1), has 27 LRR domains. There was no T-DNA insertion in AtRLP19 because mutation in this gene causes lethality [10]. All of the LRR domains in one protein form a single continuous structure and adopt an arc or horseshoe shape, a bowed tube on the concave face, a parallel β sheet, and a convex face with a variety of secondary structures, such as α helices, 310 helices, pII, and β turns [23]. LRR domains forms a stack of the parallel β strands on the inner concave face in both OsLRR-RLP2 and AtRLP19 (Figure 2C). A secondary helical structure is located on the outer convex face. In contrast to OsLRR-RLP2, AtRLP19 has 27 LRR domains, resulting in an expanded structural profile that includes a larger stack of parallel strands and transmembrane (TM) domains. To date, no functional studies have been conducted on this gene in any species.

### 2.2. Expression and Localization Analysis of OsLRR-RLP

Transcriptome data from *Nipponbare* (CAFRI-Rice database, https://cafri-rice.khu.ac.kr/, accessed on 21 February 2023) showed *OsLRR-RLP2* is expressed at very low levels in most tissues/organs except the leaves with a high expression (Figure 2D). To detect the gene transcripts in *japonica* cultivar Dongjin (cv. DJ), we performed quantitative reverse transcription PCR (qRT-PCR) analysis using the total RNA prepared from vegetative and reproductive organs. Similar to the RNA-seq data, expression of *OsLRR-RLP2* is overall low but preferential in the leaves (Figure 2E).

Protein localization was examined to validate the presence of the gene in the cellular level. Using online tools, the predominant site of OsLRR-RLP protein localization was predicted to be plasma membrane. Subcellular localization was observed through transient expression of the C-terminal green fluorescent protein (GFP)-tagged protein in tobacco epidermal cells. Control (*p35S:GFP*) signals were distributed in the nucleus, cytoplasm, and plasma membrane of epidermal tobacco cells (Figure 3A). OsLRR-RLP2 signals, which express the *p35S:OsLRR-RLP2-GFP* construct, were observed in the cytoplasm and plasma membrane (Figure 3B). To distinguish the signals between the plasma membrane and the cell wall, the leaves were treated with 1 M NaCl. After plasmolysis, both control and OsLRR-RLP2 GFP signals were detected in the cytoplasm and plasma membrane, but not in the cell wall, when observed via merging with FM4-64, a plasma membrane marker (Figure 3C,D). The protein localization is similar to that of OsRLP1, which localizes to the cytoplasm and plasma membrane [19].

### 2.3. Haplotype Analysis of OsLRR-RLP2 Sequence and Validation of M. oryzae Response

Using the rice SNP database, we observed that *OsLRR-RLP2* was a subspecies-unbalanced gene whose frequency in *japonica* is 5% greater than that in *indica* subgroups, indicating a *japonica*-dominant gene (Figure 4A). The allele frequency of *OsLRR-RLP2* in different rice subpopulations showed sequence variation compared to reference-type alleles, particularly in *indica* and *australia* (*aus*) (Figure 4B). The tropical and temperate rice types mostly had reference-type alleles, whereas the *indica* and *aus* groups possessed alternate alleles and belonged to another haplotypic group compared with the *japonica* type. In the *OsLRR-RLP2* gene, a total of 43 SNPs were identified. Among these, 28 were identified as nonsynonymous SNPs (Figure 4C).

To confirm the defense responses based on SNP differences, accessions from the Korean rice collection by re-sequenced whole genome (RWG) were acquired [24]. The RWGs were categorized as *temperate japonica* and *indica*. Among these, we obtained 9 RWG lines with indels in *OsLRR-RLP2* (Appendix A). To investigate the correlation between sequence variation and *M. oryzae* responses, we inoculated 9 RWG lines using punch-inoculation with *M. oryzae*. Following inoculation, compared with the DJ cultivar, susceptibility was observed in *indica* accessions, RWG-006 (CT9993-5-10-1-M, Colombia introduction rice), RWG-034 (BALA, India introduction rice), RWG-123 (Chungdo Hwayang 14, Korean weedy rice), and RWG-165 (Milyang 23, Korean breeding line), whereas resistance was observed in *japonica* accessions RWG-084 (Syalebyeo-163-1-B, Korean weedy rice), RWG-138 (Jinbu Byeo, Korean breeding line), RWG-140 (Hopyung, Korean breeding line), RWG-231 (Gangchan, Korean breeding line), and RWG-281 (Jinbaek, Korean breeding line) (Figure 5A,B). Fungal biomass was higher in susceptible lines than in resistant lines (Figure 5C). A subsequent PCR-based sequencing analysis was conducted to examine the *OsLRR-RLP2* sequences of the RWG lines. Different sequence sizes of RWG lines were observed using PCR of the 1.5 kb full-length gene in combination with primers F1 and R2 (Appendix A). Further sequencing of the PCR product using primers F2 and R2 showed deletion of the fourth LRR domain in *indica* RWG-006, RWG-034, RWG-123, and RWG-165 compared with intact LRR domains in *japonica* accessions (Appendix A). Although other possibilities of *M. oryzae* response to RWG lines exist, *japonica* varieties were resistant and *indica* varieties were susceptible to rice blast disease, to which the sequence variation in *OsLRR-RLP2* could be a contributing factor.

### 2.4. Functional Analysis of OsLRR-RLP2 Involving Rice Blast Disease

To examine the role of OsLRR-RLP2 in rice immunity against blast disease, we isolated a homozygous (HM) T-DNA-tagged mutant, 3D-01328 (Appendix A), and generated overexpression and knockout lines (Appendix A). To verify T-DNA insertion, we analyzed β-glucuronidase (GUS) activity. Histochemical staining for GUS activity in transgenic rice plants showed that *OsLRR-RLP2* was constitutively expressed, and GUS staining was observed in the main vein tissue of rice leaves (Figure 6A). We used PCR-based genotyping to confirm T-DNA insertion in the rice genome; the amplified PCR products showed the presence of an approximately 500 bp band for *OsLRR-RLP2* genes from transgenic plants, and no amplified DNA fragment from wild-type (WT) plants confirmed the insertion of T-DNA fragments into the rice genome (Appendix A). Inserted T-DNA vector is pGA2772, which contains enhancer element, therefore we examined the expression value. T-DNA transgenic plants exhibited higher *OsLRR-RLP2* expression levels than the WT plants (Appendix A), indicating it is activation tagging line.

To confirm overexpressed gene effect in activation T-DNA tagging line, we constructed overexpressing plants using the ubiquitin promoter, and *OsLRR-RLP2* transcription levels in *OsLRR-RLP2-OE* plants were ten times higher than in WT plants (Appendix A). Moreover, we generated knockout plants using the clustered regularly interspaced palindromic repeat (CRISPR)/CRISPR-associated protein 9 (Cas9) system. Through genotyping, we selected knockout *oslrr-rlp2-cas1* and *oslrr-rlp2-cas2* for further analysis, which are homozygous mutants with 7 bp deletions and 1 bp insertions, respectively, for further analysis (Appendix A).

We performed a blast disease assay on leaves after punch-inoculation with *M. oryzae*. WT, *OsLRR-RLP2-OE*, and *oslrr-rlp2-cas* plants were inoculated with *M. oryzae,* and disease symptoms were quantified at 7 days post inoculation (dpi). The lesion size on *oslrr-rlp2-cas* leaves was substantially larger than that on WT leaves, whereas that on *OsLRR-RLP2-OE* leaves was considerably smaller (Figure 6B,C). The propagation of *M. oryzae* on infected leaves was quantified via qRT-PCR using primers specific for *MoPot2*, an *M. oryzae* housekeeping gene. In agreement with the resistant phenotype, the relative fungal biomass was notably smaller in the rice plants inoculated with *OsLRR-RLP2-OE* than in the WT plants, whereas the expression was higher in *oslrr-rlp2-cas* (Figure 6D). *oslrr-rlp2-T HM* (*T-DNA HM*) lines exhibited enhanced resistance to *M. oryzae* compared with the WT (Figure 6E,F). *oslrr-rlp2-cas* showed susceptibility to *M. oryzae* compared with the WT, whereas *OsLRR-RLP2-OE* exhibited resistance compared with the WT. These results indicate that OsLRR-RLP2 positively regulates the blast resistance response in rice. PRRs initially sense pathogens via perception PTI [8]. PTI signaling includes the generation of reactive oxygen species, the activation of mitogen-activated protein kinases (MAPKs), and the expression of immune-related genes [25]. We propose that *OsLRR-RLP2* may induce PTI signaling and MAPK cascades by interacting with a co-receptor (Figure 6G).

## 3. Discussion

Global rice production must double by 2050 to meet the demands of the growing population [26]; however, rice yield is affected by biotic and abiotic stresses. The plant immune system is initially mediated by receptors; PRRs, the receptor proteins, include RLKs or RLPs, and typically have LRR domains [27]. We identified *OsLRR-RLP2*, containing six LRR domains, as a positive regulator of rice blast resistance with predominance in *japonica* rice. Overexpression of *OsLRR-RLP2* increased rice defenses, whereas the *OsLRR-RLP2* mutant was more susceptible to rice blast infection than WT plants. RLPs are plasma membrane-localized receptors that lack intracellular signaling domains.

LRR-RLPs, characterized by having an extracellular LRR domain-recognized protein, but that lack the intracellular kinase domain, are involved in signal transduction [13,28]. The RLP genes in several plant species have been implicated in disease resistance. RXEG1, an RLP from *N. benthamiana*, is associated with XEG1 via the LRR domain in the apoplast and forms a complex with the LRR-receptor-like kinases BAK1 and SOBIR1 to transduce XEG1-induced defense signals [29]. Arabidopsis RLP1 and AtRLP1 have also been implicated in PAMP recognition and PTI activation [17]. Despite AtRLP1 lacking glycine residues in its TM, AtRLP1 is capable of binding to the coreceptor RLK, named AtSOBIR1 [30]. Another RLP, AtRLP23, mediates the recognition of a peptide motif (nlp20) found in numerous bacterial, fungal, and oomycete necrosis- and ethylene-inducing peptide 1-like proteins (NLPs) [30]. AtRLP30 plays a role in resistance against necrotrophic fungi in Arabidopsis [31]. PRRs are involved in PTI signaling and MAPK signaling by the recognition of PAMPs [32]. Rice receptor-like kinase Xa21, a PRR for *Xanthomonas oryzae* pv. oryzae (*Xoo*), revealed PTI defense, recognizing *Xoo* Ax21 involved in PAMPs [33]. OsRLCK185 activates MAPK signaling in response to PAMPs by OsCNGC9 [34]. OsCERK1 activates MAPKs through phosphorylation OsRLKC185 and chitin-induced ROS burst [35]. RLPs associate with BAK1 and CERK1, which function as signaling kinase domain activated upon ligand perception [36,37]. In Arabidopsis, rice, and tomato, the LRR-RLK SOBIRs were found to associate with LRR-RLP, ReMAX, OsRLP1, and Cf-4, respectively, and these RLK/RLP complexes mediate immune responses [19,38,39]. Further investigations to identify the adaptor kinases associated with OsLRR-RLP2 and the recognition of *M. oryzae* PAMP, such as MSP1 [40], will reveal the mechanism of RLP in the rice immune response.

Understanding the subcellular localization of resistance genes is important because of the site of interaction between R proteins and cognate pathogen effector proteins. Protein localization and molecular functions can be determined through transient expression in transgenic plants. R proteins are typically localized in the cytoplasm, nucleus, or cell membrane [41,42]. The cytoplasmic localization of resistance proteins contributes to the hypersensitive response (HR) [43]. In the future, whether *OsLRR-RLP2* is required to induce HR will need to be determined. We also performed a GUS assay to determine the insertion of T-DNA and expression pattern of *OsLRR-RLP2*. The expression of this gene in the leaves was high in the main vein tissue. In previous researches, several LRR-RLKs are predominantly expressed in leaf vascular tissues [44]. Plant vascular bundles are responsible for the transport of water and nutrients and are colonized by *M. oryzae* [45]. Therefore, expression in the vasculature indicates that *OsLRR-RLP2* functions there to provide resistance against *M. oryzae*. This suggests that *OsLRR-RLP2* plays a role in the immune response in leaf veins.

Genome-wide analyses of the LRR-RLP family have been conducted in Arabidopsis, rice, *Brassica napus*, and poplar [10,16,46,47]. Multicopy tandem repeats support the diversifying selection of *R* genes available for adaptation to pathogen challenges [16]. The *R* genes *Pid3*, *Pi5*, *Pb1*, and *Pik* are specifically distributed in the genomes of rice subspecies, *japonica*-type accessions, and *indica*-type accessions [48]. These results are possible because *R* genes have experienced substantial differentiation owing to reproductive isolation during the evolutionary process. This distribution specificity of *R* genes implies that the *R* genes conferring resistance to *M. oryzae* differ between the *indica* and *japonica* subspecies. The SNP-seek database revealed that *OsLRR-RLP2* has 43 SNPs, 8 deletions, and 3 insertions in exons. However, *OsLRR-RLP2* has more deletions in the fourth LRR domain. This variation can be attributed to distinctions arising from the Korean landrace compared to others. The results of *M. oryzae* inoculation analysis indicated that the *indica* group with partial deletion of the *OsLRR-RLP2* showed susceptibility, and the *japonica* group with an intact *OsLRR-RLP2* showed resistance. However, in some lines, mutations in the *OsLRR-RLP2* domain did not completely match the phenotypic characteristics of rice blast disease. Other mechanisms may also occur in RWG lines. Use of the *OsLRR-RLP2* gene shows considerable potential for advancing research on blast disease resistance in *japonica* rice.

## 4. Materials and Methods

### 4.1. Plant Materials and Blast Inoculation

All the rice plants including WT, transgenic plants, and RWG lines were grown in a growth chamber at 28 °C with a 12/12 h (light/dark) photoperiod. To obtain seeds, rice plants were grown in a greenhouse at 28 °C (16 h light) and 22 °C (8 h dark), or in a living modified organism (LMO)-regulated paddy field at Pusan National University in Miryang, or at Kyung Hee University in Suwon, Republic of Korea. The seeds were germinated on MSO medium for approximately 10 days and then transferred to soil. Approximately 4- to 5-week-old seedlings were used for *M. oryzae* PO6-6 inoculation. The uppermost 2 to 4 expanded rice leaves were inoculated with *M. oryzae*. The disease response was recorded at 7 dpi by measuring the lesion length of the infected leaves, based on which the mean and standard deviation were calculated.

The RWG lines were selected from *Oryza sativa* varieties with the *LRR-RLP* gene with many mutations through insertions/deletions (indels) analysis. Consequently, 9 RWG lines were selected, which are listed in Appendix A. Line 3D-01328 carrying a T-DNA-GUS insertion in Os02g39660 (cv. Dongjin) was detected in a T-DNA insertion mutant population [49].

### 4.2. Gene Identification and Sequence Analysis

We identified OsLRR-RLP2 proteins from the Rice Genome Annotation Project (RGAP, http://rice.uga.edu/, accessed on 21 February 2022) using a Pfam domain search. We researched *OsLRR-RLP2* using CAFRI-Rice (https://cafri-rice.khu.ac.kr/, accessed on 21 February 2022). To construct the phylogenetic tree, we collected protein sequences using NCBI blast (https://blast.ncbi.nlm.nih.gov/Blast.cgi, accessed on 13 June 2022) and Phytozome (https://phytozome-next.jgi.doe.gov/, accessed on 13 June 2022). Phylogenetic analysis was performed using the neighbor-joining method with 500 bootstrap repeats in MEGA4. Three-dimensional (3D) models were predicted using OsLRR-RLP2 and AtRLP19 as templates in AlphaFold (https://alphafold.ebi.ac.uk/, accessed on 13 July 2022).

SNP and haplotype variations were confirmed using the SNP-Seek database (https://snp-seek.irri.org/, accessed on 26 November 2022). An allele frequency chart was retrieved from Rice Pan-genome Browser (https://cgm.sjtu.edu.cn/3kricedb/index.php, accessed on 26 November 2022).

### 4.3. Construction of Vectors and Plant Transformation

To construct the plasmids, a guide RNA for the CRISPR/Cas9 vector was designed, and two target regions were selected using CRISPRdirect software (http://crispr.dbcls.jp/, accessed on 1 December 2021). The gRNA fragment with annealed primers was ligated into the cloning site of the pRGEB32 binary vector [50]. In addition, the overexpression lines were cloned using *OsLRR-RLP2* CDS and *Ubiquitin* promoter into pGA3426 fusion vector. The vectors were transformed into *Escherichia coli* TOP10, and the verified plasmid was transformed into *Agrobacterium tumefaciens* LBA4404 for rice transformation. Rice callus were cocultured with *Agrobacterium* to generate transgenic rice [51].

### 4.4. DNA Isolation and Genotypic Analysis

Rice leaves were sampled for genomic DNA extraction using the classical cetyltrimethylammonium bromide (CTAB) extraction buffer. gDNA was used to sequence the CRISPR/cas9 mutants and RWG lines. A PCR-based genotyping assay was performed to screen for homozygous transgenic lines and genotype SNPs. The primers used are listed in Appendix A. In addition, fusion transcript T-DNA-tagged genes were detected via amplified PCR using the synthesized DNA as the template, GUS primers, and gene-specific primers (Appendix A).

### 4.5. RNA Isolation, cDNA Synthesis, and qRT-PCR

Total RNA was extracted using the leaf tissue infected with *M. oryzae* strains with an RNeasy Plant Mini kit (Qiagen, Hilden, Germany). cDNA was synthesized from RNA (Takara, Shiga, Japan). qRT-PCR analysis was performed using a Roter-Gene Q system (Qiagen, Hilden, Germany) and 2X SYBR Master mix. Relative expression was calculated with qRT-PCR using the threshold cycle value (CT) of each target gene against the CT value of the rice genomic *ubiquitin* (*OsUbi*) gene according the 2^−ΔΔCT^ method [52]. qRT-PCR was conducted with three replications. The primer sequences used for qRT-PCR analysis are listed in Appendix A.

### 4.6. Generation of Fusion Proteins and Subcellular Localization Analysis

For in silico analysis, online software, Plant-mSubP (https://bioinfo.usu.edu/Plant-mSubP/, accessed on 6 January 2023) and LocTree3 (https://rostlab.org/services/loctree3/, accessed on 6 January 2023), was used to determine the subcellular localization of OsLRR-RLP2 protein.

*Nicotiana benthamiana* localization assays were performed as described by Sparkes et al. [53]. The entire coding region of *OsLRR-RLP2* was amplified using primers LRR-RLP_F and LRR-RLP_R (Appendix A). Validated cDNA inserts were cloned into the *p35S:OsLRR-RLP2-GFP* fusion vector at the 5′-end of the *GFP* gene via digestion with HindIII and EcoRI. The resulting fusion construct and control vector (*p35S:GFP*) were transformed into *A. tumefaciens* strain GV3101. Transformed GV3101 was infiltrated into *N. benthamiana* leaves. Transient expression in *N. benthamiana* leaves was observed using a K1-Fluo confocal microscope (Nanoscope System, Daejeon, Republic of Korea) after 72 h of infiltration and observed with GFP at 480 nm excitation and 550 nm emission wavelengths.

To detect the membrane localization of OsLRR-RLP2, FM4-64 staining of *N. benthamiana* leaves was performed and observed with red fluorescent protein (RFP) at 400 nm excitation and 500 nm emission wavelengths.

### 4.7. Histochemical GUS Assay

Leaves were immersed in a GUS staining solution [54]. After the tissues had been incubated at 37 °C for 2 h, the chlorophyll was removed in 70% ethanol. The samples were photographed using a BX21 microscope (Olympus, Tokyo, Japan).

## 5. Conclusions

RLK and RLP activate the defense response by recognition of PAMP as the PRR signaling. RLK and RLP play a critical role in plant immunity, growth, and development, but the mechanisms of RLP, which lack an intracellular signaling kinase domain, are not as well understood as those of RLK. We identified that *OsLRR-RLP2* is implied to be a positive regulator to rice blast disease in *japonica* rice. Additionally, *OsLRR-RLP2* has differences in the DNA sequence in the rice accessions, as genetic differentiation was occurred in the *R* gene. We observed that *japonica* varieties with intact *OsLRR-RLP2* showed resistance. Our results suggest that the *OsLRR-RLP2* gene can be used for research on blast disease resistance in *japonica* rice.

## Figures and Tables

**Figure 1 ijms-25-02216-f001:**
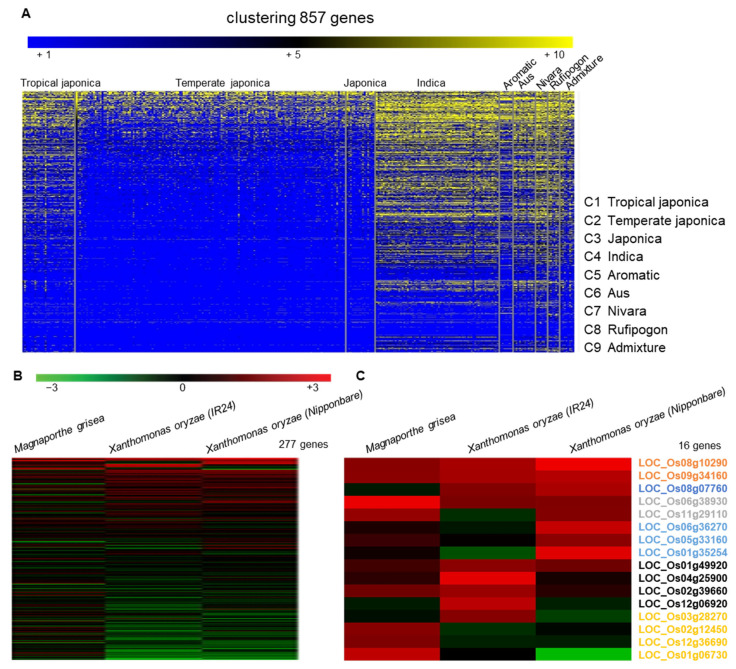
Heatmap analysis for indel (≥3 bp) frequency of 453 high quality accessions compared to *Nipponbare* of 857 leucine-rich repeat domain (LRR) genes in rice. (**A**) Heatmap representing genes with LRR domains between difference species, (**B**) 277 genes significantly differentially expressed under biotic stress, and (**C**) 16 genes were upregulated under blast and white bacterial blight disease treatments.

**Figure 2 ijms-25-02216-f002:**
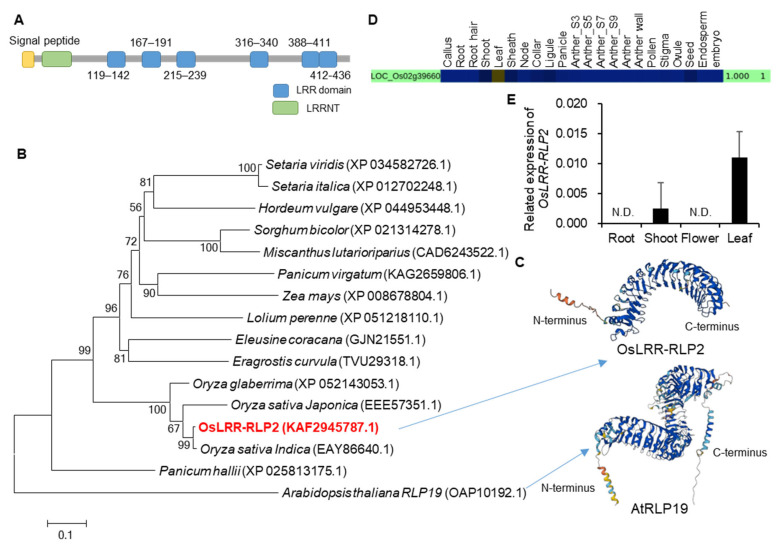
Results of protein domain, phylogenetic tree, and gene expressions analyses. (**A**) Schematic representation of *OsLRR-RLP2* using SMART (http://smart.embl-heidelberg.de/, accessed on 21 February 2022). LRR, leucine-rich repeat domain; LRRNT, leucine-rich repeat N-terminus. (**B**) Phylogenetic tree of *OsLRR-RLP2*. *OsLRR-RLP2* is in red font. The protein sequences were identified using NCBI BLASTX. Phylogenetic tree based on the neighbor-joining method was constructed using MEGA 4 with 500 bootstrap replications. (**C**) Three-dimensional modeling of OsLRR-RLP2 and AtRLP19 protein. (**D**) CAFRI-RICE analysis of *OsLRR-RLP2*. *OsLRR-RLP2* showed high expression in leaves. (**E**) Validation of *OsLRR-RLP2* expression using qRT-PCR analysis. The roots and shoots of 10-day-old seedlings, flowers, and 3-month-old leaves were analyzed. Three technical replicates were performed, and the results were normalized using rice *ubiquitin* (*OsUbi5*, Os01g22490). N.D., Not detection.

**Figure 3 ijms-25-02216-f003:**
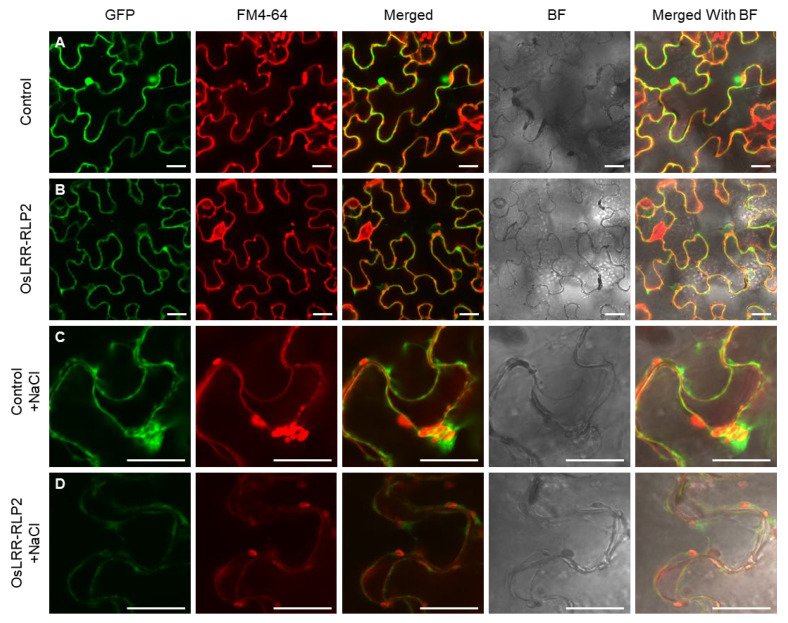
Subcellular localization of OsLRR-RLP2. GFP signals of *OsLRR-RLP2* in tobacco epidermal cells stained with the membrane marker FM4-64. The tobacco leaves were treated with 1 M NaCl to induce plasmolysis. (**A**,**C**) Control vector (*p35S:GFP*) expressed in epidermal cells of tobacco leaves. (**B**,**D**) Recombinant vector (*p35S:OsLRR-RLP2-GFP*) expressed in epidermal cells of tobacco leaves. GFP, green fluorescent protein; BF, bright-field; scale bar: 30 μm.

**Figure 4 ijms-25-02216-f004:**
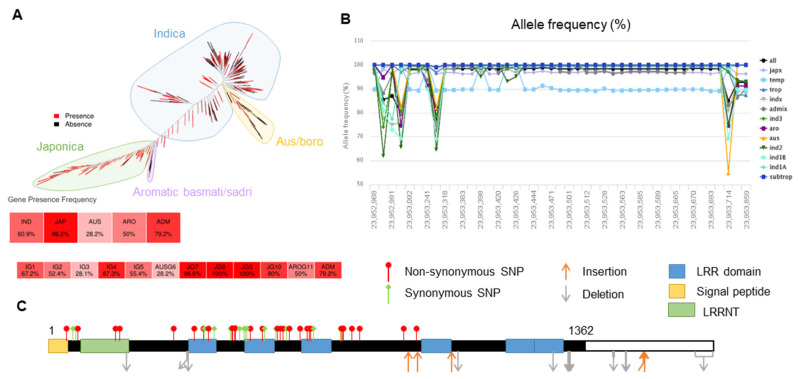
Single-nucelotide polymorphisms (SNPs) and synteny of the genomic sequences of *OsLRR-RLP2*. (**A**) Phylogenetic tree constructed based on presence/absence variations (PAVs) among 453 high quality accessions. *OsLRR-RLP2* can be classified into *japonica*-dominant genes and *Indica*-subgroup-unbalanced genes. (**B**) Allele frequencies of *OsLRR-RLP2* in 3024 accessions available from SNP-Seek database. Numbers on X-axis represent positions of indels. Numbers on Y-axis indicate allele frequency in percentage. (**C**) SNPs of *OsLRR-RLP2* gene across the ~3024 rice accessions depicted based on the sequence variation data retrieved from the SNP-Seek database.

**Figure 5 ijms-25-02216-f005:**
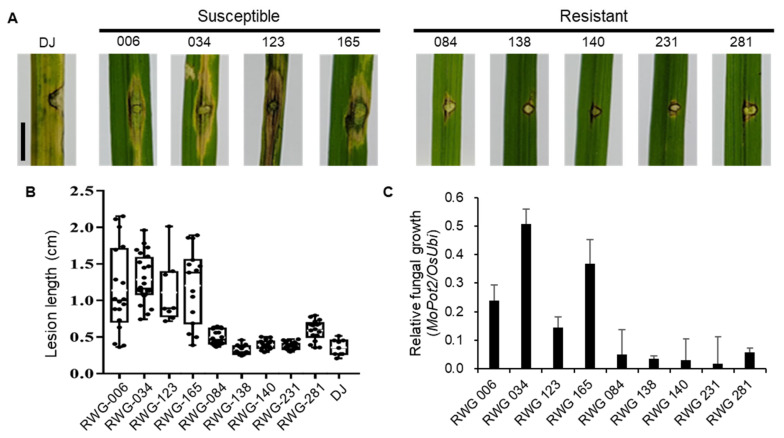
Inoculation of DJ and 9 RWG lines in rice blast disease. (**A**) Representative leaves showing disease lesions in five-week-old from RWG lines inoculated with *M. oryzae* (PO6-6). The leaves were photographed 7 days post inoculation (dpi). Scale bar = 1 cm. (**B**) Disease lesion lengths in leaves of five-week-old RWG lines were measured 7 dpi. (**C**) Relative fungal biomass determined by examining the expression level of *M. oryzae Pot2* against *OsUbiquitin* DNA level. Three technical replicates were performed. Error bars indicate ± SD.

**Figure 6 ijms-25-02216-f006:**
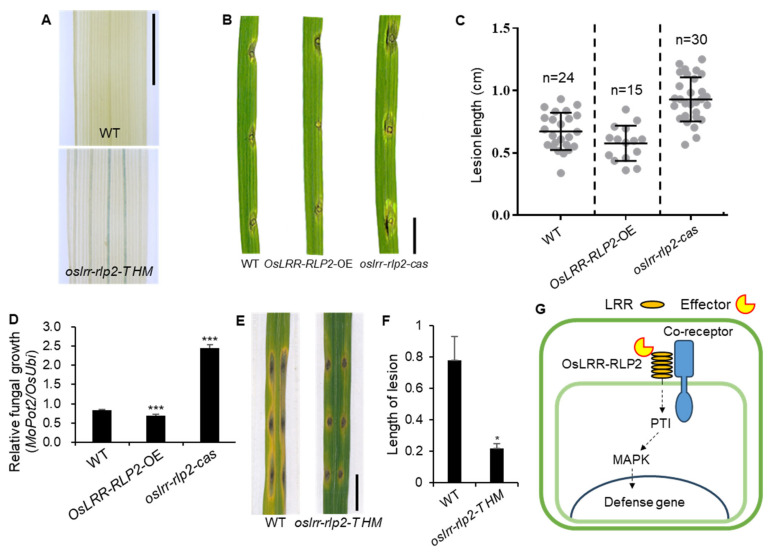
Functional analysis of *OsLRR-RLP2*. (**A**) GUS staining of T-DNA plants. Expression appears in main vein tissue of the leaf. Scale bar = 1 cm. (**B**) Representative leaves from WT, *OsLRR-RLP2*-OE, and *oslrr-rlp2-cas* inoculated with *M. oryzae* (PO6-6). The leaves were photographed at 7 dpi. Scale bar = 1 cm. WT, Wild-type; OE, overexpression. (**C**) Disease lesion lengths in five-week-old leaves from WT and transgenic plants were measured at 7 dpi with *M. oryzae* (PO6-6). (**D**) Relative fungal was determined by examining the expression levels of *M. oryzae Pot2* and *OsUbiquitin* DNA. Three technical replicates were performed. Error bars indicate ± SD. Asterisks indicate significant differences between samples according to the *t*-test (*** *p* < 0.001). (**E**) Representative leaves from WT and *oslrr-rlp2 T-DNA HM* inoculated with *M. oryzae* (PO6-6). The leaves were photographed at 7 dpi. Scale bar = 1 cm. (**F**) Disease lesion lengths in five-week-old leaves from WT and *oslrr-rlp2-t HM* were measured at 7 dpi with *M. oryzae* (PO6-6). Asterisks indicate significant differences between samples according to the *t*-test (* *p* < 0.05). (**G**) Model for defense activation by OsLRR-RLP2.

## Data Availability

Data are contained within the article.

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
