# Peer review of "OsLRR-RLP2 Gene Regulates Immunity to Magnaporthe oryzae in Japonica Rice"

_ijms, 2024, doi:10.3390/ijms25042216_

Round 1

Reviewer 1 Report

Comments and Suggestions for Authors

The paper “OsLRR-RLP2 Gene Regulates Immunity to Magnaporthe oryzae in Japonica Rice” identified a blast-resistance gene OsLRR-RLP2 through bioinformatics analysis. Gene knockout and overexpressing indicated that OsLRR-RLP2 conferred blast-resistance to rice, and suggested OsLRR-RLP2 may be useful for breeding resistant cultivars. I only have some small suggestions for revision to this article, as follows:

1. line 96-97: How this gene was screened through the analysis of T-DNA mutant lines needs to be clarified.

2. line 182: I dont understand why the abbreviation for Korean rice collection is RWG, please verify.

3. This article did not provide sufficient discussion on the results obtained from the research. Please add some more in-depth discussion content. 

Author Response

Response 1: According to reviewer comments, we included the information about screening analysis of T-DNA mutant lines. Please see line 100-102: “To verify the correlation between 16 genes and blast disease, we screened T-DNA-tagged mutant lines and conducted PCR-based genotyping for identification of tagging in the each of genes.

Response 2: Some of the RWG are “Korean rice collection” (Kim, Tae-Sung, et al. "Genome-wide resequencing of KRICE_CORE reveals their potential for future breeding, as well as functional and evolutionary studies in the post-genomic era." BMC genomics 17 (2016): 1-13.). From this study, ‘RWG’ was named in the process of securing the species by confirming genotype information through Re-sequencing of the Whole Genome. We added it.

Response 3: Thank you for suggestion. We revised the result and discussion content. Please see page 9, line 289-300.

Reviewer 2 Report

Comments and Suggestions for Authors

The manuscript “OsLRR-RLP2 Gene Regulates Immunity to Magnaporthe oryzae in Japonica Rice” identified and studied the functionality of R gene, OsLRR-RLP2 through knock-out of the gene shows increased susceptibility and overexpression of it show enhanced resistance to blast. Further, subcellular localization and GUS assay were also performed in this study.

Other concerns

  1. Are there any phenotypes among the T-DNA-tagged and CRISPR mutant lines? It seems to me the T-DNA-tagged has some developmental defects, is there any yield penalty for the T-DNA-tagged and CRISPR knockout line?

  1. Did the authors try to examine any Xanthomonas oryzae pv. oryzae (Xoo) inoculation for the overexpression of OsLRR-RLP2 and the CRISPR knock-out line?

Comments on the Quality of English Language

Minor errors 

Page 3, Line 105, white bacterial blight disease, remove white

Page 7, Line 207, Relative fungal biomass was determined

Author Response

Response 1: T-DNA-tagged line delay development. Both T-DNA-tagged and CRISPR knockout line show slight decrease in fertility.

Response 2: We inoculated Magnaporthe oryzae PO6-6 (Mo) inoculation for the overexpression of OsLRR-RLP2 and the CRISPR knockout line. But we did not test Xanthomonas. Sorry to not give more information. In this study, we focused on BLAST disease.

Reviewer 3 Report

Comments and Suggestions for Authors

The objective of the study is to investigate the resistance to Magnaporthe oryzae in Japonica Rice, focusing on LRR-RLPs mechanisms. The RLPs have rarely been investigated for these aims, and the research in rice is scarce, thus the motivations of the study are appropriate respect to the state of the art.

The study is well structured and each section is adequately discussed.

Figures and tables are clear and comprehensive.

However, some minor revisions are suggested.

·         In the introduction, please add initially some sentences (two, three sentences) describing the rice blast disease and its effects on the plant, with the associated references.

·         Reference missing- Line 55

·         The sentence ”Relative fungal biomass w determined…..” contains a mistake, please modify – Line 207

·         The ‘Discussion’ section contains sentences dispersive and disorganized and at the same time already wrote in the ‘Introduction’ section. I suggest to delete sentences from Line 261 to 265. Furthermore, many sentences appear unrelated from the other parts. Authors are invited to link, to correlated better the sentences to each other.

·         The ‘Conclusions’ section is absent. I suggest to include it underlying the main achieved goals.

Furthermore, please, pay attention to the format of some words and of the manuscript.

·         For instance, check the Italic form for the OsLRR-RLP2 gene - Lines 158, 246

·         Table of abbreviations at the end of the document is suggested

·         Please check the reference format (especially about volume and issue number)

Comments on the Quality of English Language

Minor editing of English language required

Author Response

Response 1: Thank you for comments.We edited statements in the manuscript about the rice blast disease and its effects on the plant with the associated references (please see line 34-38).

Response 2: We added a missing reference [14]:

Hohmann, U.; Lau, K.; Hothorn, M., The structural basis of ligand perception and signal activation by receptor kinases. Annual review of plant biology 2017, 68, 109-137.

Response 3: We edited the word (please see page 7, line 213).

Response 4: Thank you for the comments. We removed some overlapping sentences and edited the discussion section (please see page 8, line 270-275).

Response 5: During revision, we added the ‘5. Conclusions’ section. Please see page 11, line 411-420:

“5. Conclusions

RLK and RLP activate the defense response by recognition of PAMP as the PRR signaling. RLK and RLP play a critical role in plant immunity, growth, and develop-ment, but the mechanisms of RLP, which lack an intracellular signaling kinase domain, are not as well understood as those of RLK. We identified that OsLRR-RLP2 is implied to be a positive regulator to rice blast disease in japonica rice. Additionally, OsLRR-RLP2 has differences in the DNA sequence in the rice accessions, as genetic differentiation was occurred in the R gene. We observed that japonica varieties with in-tact OsLRR-RLP2 showed resistance. Our results suggest that the OsLRR-RLP2 gene can be used for research on blast disease resistance in japonica rice.

Response 6: We italicized the OsLRR-RLP2 gene. Please see page 5, line 164 and page 8, line 257.

Response 7: We added the table of abbreviations in revised manuscript. Please see page 12.

Response 8: We checked the reference format and edited them.